# Inflammation promotes resistance to immune checkpoint inhibitors in high microsatellite instability colorectal cancer

Qiaoqi Sui [1,2,3,7], Xi Zhang[4,7], Chao Chen[5,7], Jinghua Tang [1,2,3,7], Jiehai Yu[1,2,3,7], Weihao Li[1,2,3,7], Kai Han [1,2,3], Wu Jiang[1,2,3], Leen Liao[1,2,3], Lingheng Kong[1,2,3], Yuan Li[1,2,3], Zhenlin Hou [1,2,3], Chi Zhou[1,2,3], Chenzhi Zhang[1,2,3], Linjie Zhang[1,2,3], Binyi Xiao[1,2,3], Weijian Mei [1,2,3], Yanbo Xu[1,2,3], Jiayi Qin[2,3,6], Jian Zheng [2,3,6], Zhizhong Pan[1,2,3] & Pei-Rong Ding [1,2,3] ✉

Inflammation is a common medical complication in colorectal cancer (CRC) patients, which plays significant roles in tumor progression and immunosuppression. However, the influence of inflammatory conditions on the tumor response to immune checkpoint inhibitors (ICI) is incompletely understood. Here we show that in a patient with high microsatellite instability (MSI-H) CRC and a local inflammatory condition, the primary tumor progresses but its liver metastasis regresses upon Pembrolizumab treatment. In silico investigation prompted by this observation confirms correlation between inflammatory conditions and poor tumor response to PD-1 blockade in MSI-H CRCs, which is further validated in a cohort of 62 patients retrospectively enrolled to our study. Inhibition of local but not systemic immune response is verified in cultures of paired T cells and organoid cells from patients. Single-cell RNA sequencing suggests involvement of neutrophil leukocytes via CD80/CD86-CTLA4 signaling in the suppressive immune microenvironment. In concordance with this finding, elevated neutrophil-to-lymphocyte ratio indicates inhibited immune status and poor tumor response to ICIs. Receiver operating characteristic curve further demonstrates that both inflammatory conditions and a high NLR could predict a poor response to ICIs in MSI- CRCs, and the predictive value could be further increased when these two predictors are combined. Our study thus suggests that inflammatory conditions in MSI-H CRCs correlate with resistance to ICIs through neutrophil leukocyte associated immunosuppression and proposes both inflammatory conditions and high neutrophil-to-lymphocyte ratio as clinical features for poor ICI response.

High microsatellite instability (MSI-H), highly correlated with DNA mismatch repair deficiency (dMMR), plays a prominent role in the tumorigenesis of colorectal cancer (CRC)[1,2]. MSI-H and dMMR are associated with high mutation burden, high tumor neoantigen load and dense infiltration of immune cells[3,4]. It has been well accepted that a dMMR/MSI-H status benefits CRC patients receiving immune checkpoint inhibitors (ICI), especially PD-1 blockade[5,6]. However, recent studies have reported that the immune status differs among

dMMR/MSI-H CRCs, and over 50% of patients still experience resistance to ICIs with an unclarified mechanism[6].

Inflammatory conditions caused by obstruction or perforation are common complications in CRCs[7]. It has been revealed that both local and systemic inflammation have important roles in tumorigenesis, disease progression, and patient prognosis in various cancers[8–10]. Recent studies have found that local inflammation is also associated with immunosuppression, and that elevated inflammatory cells in the tumor microenvironment are associated with resistance to ICIs[11–13]. In addition, the inflammatory response is associated with alterations in peripheral blood leukocytes that can be captured by a high neutrophil-to-lymphocyte ratio (NLR), which is also associated with poor long-term survival across all ICIs in patients with various soild tumors[14–16]. However, the significance of inflammation in the response to ICIs remains unclarified in MSI-H CRCs.

Here, the current study reveals that local inflammatory conditions impair tumor response to ICIs among MSI-H CRC patients. We also demonstrate that increased neutrophil infiltration correlates to inhibited tumor immune status, and that the activated CD80/CD86-CTLA4 axis in inflammation participates in the T cell exhaustion. Moreover, our data indicate that both inflammatory conditions and an NLR > 3 are potential predictors of poor response to ICIs clinically. These findings suggest an inhibitory role of inflammation in tumor response to ICIs through neutrophil-associated immune suppression, providing new clinical features and therapeutic targets for MSI-H CRCs resistant to immunotherapy.

## Results

### Inflammatory conditions in tumor site are associated with resistance to ICIs in MSI-H CRCs

A 34-year-old woman with metastatic MSI-H descending colon cancer, who had peritonitis due to perforation of primary tumor and received transverse colostomy, was enrolled (Patient 1). The primary tumor and liver metastatic lesions regressed after 3 courses of Pembrolizumab in the combination of chemotherapy. After 5 courses of treatment, the patient had a fever, with increased white blood cell count and C-reactive protein concentration. She was considered having immune-related adverse reactions, and received only chemotherapy at the sixth course. After 6 courses of treatment, the primary tumor was found progressing with elevated tumor markers, and recurrent fever remained. The patient was then considered having localized infection in the perforating site and received another 3 courses of Pembrolizumab treatment combined with anti-infectious therapy (Supplementary Fig. 1A, B, and C). The primary tumor continued to progress. Meanwhile, the metastatic lesions continually regressed after 9 courses of treatment (Fig. 1A). The patient finally received resection of the primary tumor and 15 courses of postoperative Pembrolizumab. The liver metastases were evaluated as complete remission and remained no evidence of diseases on last follow-up in July 2021.

Patient 1 achieved mixed response in the primary tumor (progressive disease, PD) and metastatic tumor (partial response, PR) after 5 courses of Pembrolizumab combined with chemotherapy and one course of chemotherapy. Localized infection around the primary tumor was noticed and considered to be one of the potential reasons leading to the resistance of primary tumor. We therefore hypothesized that local inflammatory conditions could be associated with resistance to PD-1 blockade. To demonstrate this, a cohort of 62 MSI-H CRC patients who received PD-1 blockade therapy was retrospectively included (Table 1). We found that local inflammatory conditions in tumor sites during treatment were correlated with a higher ratio of stable disease (SD) and PD (71.43% v.s. 31.25%, $P = 0.011$) and worse progression-free survival (PFS) (Fig. 1B and C). Multivariate Cox proportional hazards regression indicated that inflammatory conditions were independently associated with the risk of progression (Table 2). Besides, in 4 patients determined complete response (CR) with

inflammatory conditions, all of them received timely anti-infectious treatment.

We then constructed tumor organoids of primary tumors from 4 MSI-H CRC patients receiving PD-1 blockade (Fig. 1D and Supplementary Fig. 1D). Patient 1 had inflammatory conditions without appropriate antibiotic therapy, and had PD after PD-1 blockade therapy. Patients 2 received 4 courses of PD-1 blockade therapy with Tumor Regression Grade (TRG) 3, who had peritonitis after 2 courses of treatment and immediately received anti-infectious treatment. Patients 3 was determined TRG 3 and Patient 4 was determined TRG 2 pathologically, both of whom did not have local inflammatory conditions. Tumor organoids from Patient 1 to 4 were co-cultured with paired tumor infiltrating lymphocyte (TIL) or peripheral blood mononuclear cell (PBMC)-derived T cells. The apoptotic proportion of organoid cells was higher in the PBMC group in Patient 1, while the proportions in Patient 2, 3 and 4 were comparable between those two groups (Fig. 1E and Supplementary Fig. 2A, B). These phenomena indicate an inhibited local immune response to tumors instead of systemic immunosuppression in Patient 1.

### An inhibitory role of neutrophils in the tumor immune status is revealed by single-cell RNA sequencing (scRNA-seq)

To investigate the component of the microenvironment in the primary tumor, scRNA-seq of Patient 1, 2, 3 and 4 was conducted. In Patient 1, the 4489 qualified cells were divided into epithelial cells and immune cells (Fig. 2A, B), and Immune cells were further clustered (Supplementary Fig. 3A). Among immune cells, CD8+ T cells were mainly identified as exhausted cells expressing *PDCD1* and *KLRD1* separately (CD8 TEX PDCD1 and CD8 TEX KLRD1). CD4+ T cells were mainly regulatory T cells (Treg cells) expressing *CTLA4* (with or without high *FOXP3* expression). In addition, 132 myeloid cells were identified, which were characterized by high expression of *PTPRC, C1QB, AIF*, and *LYZ* (Supplementary Fig. 3B). Furthermore, myeloid cells expressed *ITGAM, FCGR2A, FCGR3A, FCGR3B, CD44* and *CD55*, which indicated that this cluster includes neutrophil-like cells (Fig. 2C and Supplementary Fig. 3B)[17–19]. In Patient 2, 3088 cells were identified as 8 epithelial cell clusters and 2 immune cell clusters (Supplementary Fig. 4), among which the immune cells were mostly B cells and cytotoxicity T cells expressing *GZMB* and *GNLY* (Supplementary Fig. 4). Combined with the clustering results of Patient 1, we considered that the anti-infectious treatment may improve the local immune status in patients with inflammatory conditions. In Patient 3, 2166 cells were clustered into 5 epithelial cells, Treg cell, B cell, plasma cell, CD8+ T cell, fibroblast and myeloid cell (Supplementary Fig. 5). Among myeloid cells, *FCGR2A, FCGR3A, CD44* and *CD55* were expressed, indicating that neutrophil-like cells were included (Supplementary Fig. 5). In Patient 4, 2375 cells were divided into 7 epithelial cells, plasma cell, B cell, Treg cell and fibroblast (Supplementary Fig. 6).

To visualize the neutrophil infiltration in MSI-H CRCs, immunohistochemistry (IHC) examination of CD11b (encoded by *ITGAM*) was conducted among surgically resected tissues from 21 patients. The result demonstrated that patients with poor response to ICIs had increased neutrophil infiltration (Supplementary Fig. 7). To further investigate the influence of neutrophils on immune status of MSI CRCs, The Cancer Genome Atlas (TCGA) data were analyzed using Immune Cell Abundance Identifier (ImmuCellAI). We found that predicted poor response to ICIs was associated with elevated neutrophil infiltration (Fig. 2D). Moreover, neutrophil infiltration negatively correlated with total infiltration scores (Fig. 2E) and the infiltration of cytotoxic T cells, Th1 cells, NK cells and B cells (Fig. 2F).

### Neutrophils exhaust T cells through CD80/CD86-CTLA4 axis under inflammatory conditions

To further demonstrate the interaction between immune cells, a cell-cell network analysis was conducted. In Patient 1, the result suggested

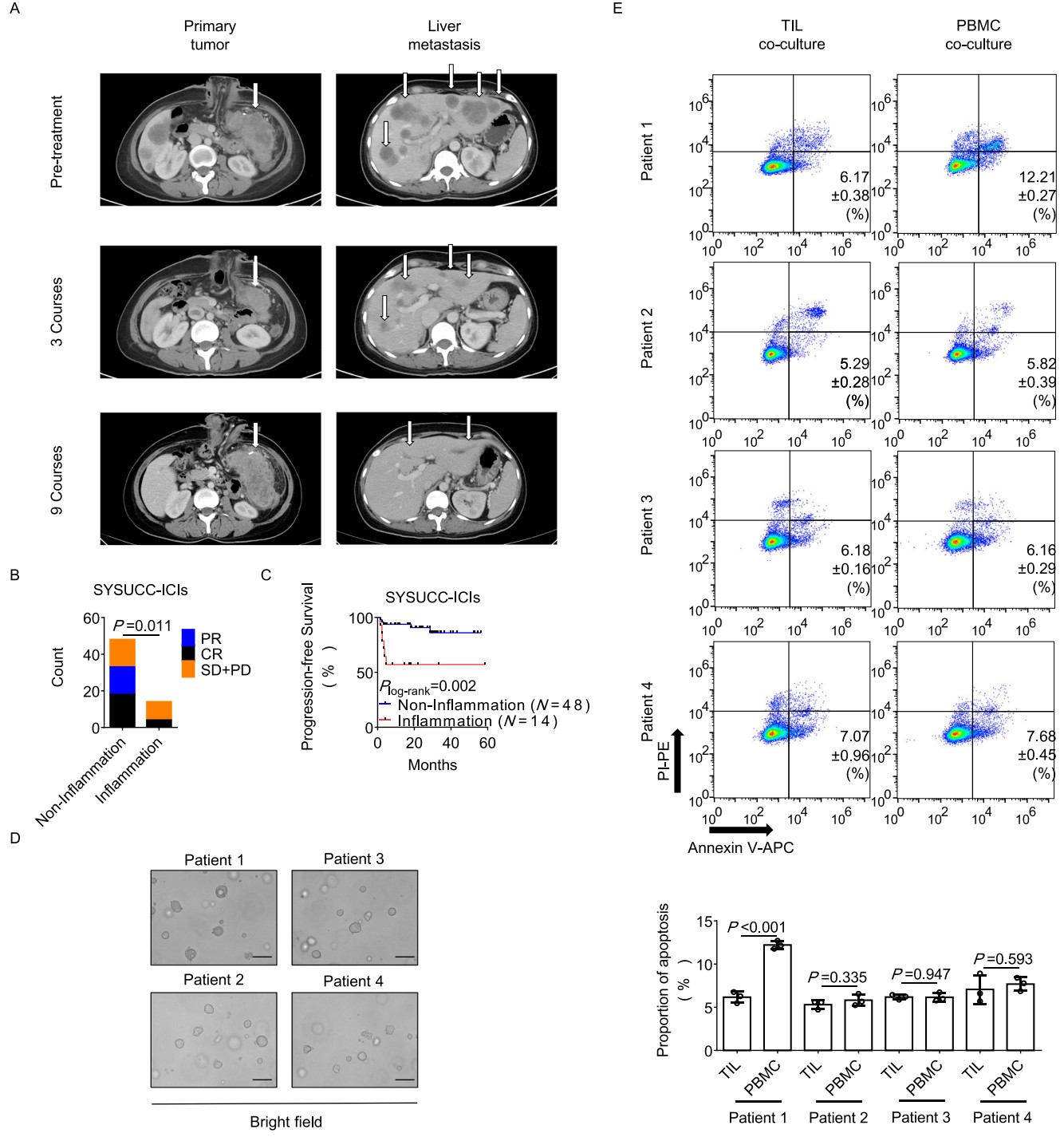

**Fig. 1 | Inflammatory conditions are associated with poor response to PD-1 blockade in dMMR/MSI-H CRCs. A** CT scans were conducted on Patient 1 after 0 courses, 3 courses and 9 courses of treatment. Images of the primary tumor and metastatic lesions are shown. Tumor lesions are marked by arrows. **B, C** Tumor response (**B**) and progression-free survival (**C**) after PD-1 blockade were compared between MSI-H CRC patients with ($N = 14$) and without ($N = 48$) inflammatory conditions. The Wald chi-square test (**B**) and Kaplan-Meier method with Log-rank test (**C**) were used. **D** Bright field (Scale bar: 100 μm) images of tumor organoids from 4 patients are shown. Each experiment was repeated 3 times. **E** Tumor organoids were co-cultured with paired TILs or PBMC-derived T cells for 6 h and were then separated for apoptosis assays. The two-tailed unpaired Student's $t$ test was used. The mean values are shown, and the standard deviations are displayed by the error bars. Each experiment was repeated 3 times with 3 replicates. Source data are provided as a Source Data file. PR partial response, CR complete response, SD stable disease, PD progressed disease, TIL tumor-infiltrating lymphocyte, PBMC peripheral blood mononuclear cell.

that myeloid cells have a core role in the immune microenvironment, as they harbored the most connections with other cell types, especially with exhausted CD8[+] T cells (Fig. 3A). In Patient 3, the importance of fibroblasts and myeloid cells ranked the top, and the fibroblasts harbored the most connections with other cells in Patient 4 (Supplementary Fig. 8A).

To investigate the interactions that occur in the ecosystem between myeloid cells and other immune cells in Patient 1 and 3, a significant ligand-receptor (L-R) pair was conducted to calculate the intensity of the interactions. In Patient 1, the analysis suggested crosstalk between myeloid cells and T cells via CD80/CD86-CTLA4 (Fig. 3B). Other L-R pairs involving chemokines and cytokines between

**Table 1 | Baseline characteristics of 62 MSI-H CRC patients receiving PD-1 blockade**

| | With inflammatory conditions | Without inflammatory conditions |
|---|---|---|
| Total | 14 | 48 |
| **Gender** | | |
| Male | 9 (64.3%) | 33 (68.8%) |
| Female | 5 (35.7%) | 15 (31.2%) |
| Median age (range) | 34 (20~73) | 46 (19~67) |
| **Family history** | | |
| Yes | 6 (42.9%) | 28 (58.3%) |
| No | 8 (57.1%) | 20 (41.7%) |
| **Histology** | | |
| Adenocarcinoma | 10 (71.4%) | 40 (83.3%) |
| Mucinous Adenocarcinoma | 4 (28.6%) | 8 (16.7%) |
| **Receiving PD-1 blockade combined with non-ICI drugs** | | |
| Yes | 9 (64.3%) | 21 (43.8%) |
| No | 5 (35.7%) | 27 (56.2%) |

*MSI-H* High microsatellite instability, *CRC* Colorectal cancer, *ICI* Immune checkpoint inhibitor.

**Table 2 | Univariate and multivariate Cox proportional hazards regression for PFS**

| Variable | No. of cases | *HR* (95% *CI*) | *P* |
|---|---|---|---|
| **Inflammatory conditions** | | | |
| No | 48 | 1.000 (ref) | – |
| Yes | 14 | 5.494 (1.652~18.273) | 0.005 |
| **Histology** | | | |
| Adenocarcinoma | 50 | 1.000 (ref) | – |
| Mucous adenocarcinoma | 12 | 4.149 (1.256~13.701) | 0.020 |
| **Family history** | | | |
| No | 28 | 1.000 (ref) | – |
| Yes | 34 | 0.286 (0.076~1.083) | 0.065 |
| **Multivariable** | | | |
| **Inflammatory conditions** | | | |
| No | | 1.000 (ref) | – |
| Yes | | 4.886 (1.445~16.520) | 0.011 |
| **Histology** | | | |
| Adenocarcinoma | | 1.000 (ref) | – |
| Mucous adenocarcinoma | | 3.539 (1.051~11.918) | 0.041 |

The statistical tests were two-sided, and variable with *P* value <0.05 was included in multivariate Cox proportional hazards regression. No adjustments were made.
*PFS* Progression-free survival, *HR* Hazard ratio, *CI* Confidence interval.

myeloid cells and T cells were also identified, including CCL5-CCR1, CCL5-CCR5, CCL3-CCR5, TNF-TNFRSF1B, IFNG-type II IFNR, and CSF1R-CSF1 (Supplementary Fig. 8B and C). Among these immune-associated ligands and receptors, CTLA4, CCR5 and CSF1R could contribute to tumor immune suppression[20–22], while TNFRSF1B and type II IFNR are associated with stimulation in neutrophils[23,24]. In Patient 3, L-R pair of CD86-CTLA4 was also found between myeloid cells and Treg cells (Supplementary Fig. 8E).

It has been reported that N2-neutrophils play a more significant role than N1-neutrophils in modulating the growth of primary tumors and progression to metastatic disease[25]. Therefore, we further clustered the myeloid cells by N1/N2-associated markers (*FAS, NOS2* and *CCL3* for N1; *MRC1, CCL2* and *ARG2* for N2) (Fig. 3C and Supplementary Fig. 9) in Patient 1[26]. The transcriptome profile suggested that the N2-cluster expressed higher level of activated neutrophil markers *ITGAM, FCGR2A, FCGR3A, FCGR3B, CD44, CD55* and *TNFRSF1B*, which indicated that N2-neutrophils may participate more in the inflammatory function (Supplementary Fig. 9). Besides, the expression level of *CD80, CD86, CCL5*, and *CSF1R* was much higher in cells of N2 cluster, which indicated that N2-neutrophils participate more in the immune suppression (Fig. 3D). To further investigate the relationship between neutrophil infiltration and L-R pair interaction occurred in Patient 1, the expressions of immune-associated ligands and receptors, as well as *ITGAM*, the marker of inflammatory-activated neutrophil, were analyzed in MSI CRCs from TCGA. The correlation assays demonstrated that the expression of *ITGAM* is positively correlated to the expression of *CD80, CD86, CTLA4, CSF1, CSF1R, CCL5, CCR1, CCR5* and *CXCL9* (Supplementary Fig. 10).

Due to that CD80/CD86-CTLA4 axis were involved in the crosstalk between myeloid cells and T cells, and that the application of CTLA4 blockade has been widespread in tumor immunotherapy, we investigated the value of targeting CD80/CD86-CTLA4 axis under inflammatory conditions. PD-1 neutralized murine T cells were stimulated in the presence or absence of neutrophils before co-cultured with murine CRC cells MC38. The apoptosis assays suggested that coculturing with neutrophils attenuated the apoptotic proportion of CRC cells, while this effect could be rescued by neutralization of CD80 and CD86 (Fig. 3E and Supplementary Fig. 8D). These results indicate that blockade of CD80/CD86-CTLA4 axis could improve the tumor response to PD-1 blockade under inflammations.

## An elevated NLR is associated with a poor immune status and resistance to ICIs in dMMR/MSI-H CRC

According to a previous study, the NLR shows a positive correlation with infiltrating neutrophils in tumors[27]. Moreover, Patient 1, 2, and 3 had NLR > 3 during their treatment, among whom Patient 1 and 2 had local inflammatory conditions. Meanwhile, inflammatory cell infiltration was observed in scRNA seq data in Patient 1 and 3 (Fig. 2A and Supplementary Fig. 5), which also suggested that NLR > 3 could be associated with elevated neutrophil infiltration. Since neutrophil infiltration is associated to poor immune status and resistance to PD-1 blockade, we hypothesized that testing NLR could also predict tumor immune status. In a cohort of 142 surgically resected dMMR/MSI-H CRCs[28], we demonstrated that the CD8+ TILs in the invasive margin (IM) were significantly lower in patients with preoperative NLR > 3 (Fig. 4A). Besides, to demonstrate that the NLR could predict the tumor response to PD-1 blockade, clinical data from the 58 MSI-H CRCs (4 without consistent blood test data) were analyzed. We found that patients with NLR > 3 were more likely to have inflammatory conditions (45.00% v.s. 10.53%, *P* = 0.003, Fig. 4B) and higher SD + PD ratio (70.00% v.s. 21.05%, *P* = 0.001, Fig. 4C). These results suggest that an NLR > 3 is associated with poor immune status and poor response to PD-1 blockade in MSI-H CRCs.

To compare the efficiency between the NLR and local inflammatory conditions in predicting poor tumor response (SD + PD) to ICIs, Receiver operating characteristic (ROC) curves were used. The areas under the curve (AUC) of having an inflammatory condition was 0.649 [*P* = 0.059, 95% confidence intervals (*CI*), 0.496-0.802], while the AUC of having an NLR > 3 was 0.735 (*P* = 0.003, 95% *CI*, 0.595-0.875). When two predictors were combined, the AUC was 0.771 (*P* = 0.001, 95% *CI*, 0.639-0.904) (Fig. 4D). These results demonstrate that both inflammatory conditions and a high NLR could predict a poor response to ICIs in MSI-H CRCs, and the predictive value could be further increased when these two predictors are combined.

## Discussion

ICIs are very effective treatments for patients diagnosed MSI-H CRCs, but 30–50% of those patients present primary or secondary resistance to the treatment[5,29]. In the present study, we demonstrate that MSI-H CRC patients with inflammatory conditions have higher risks of

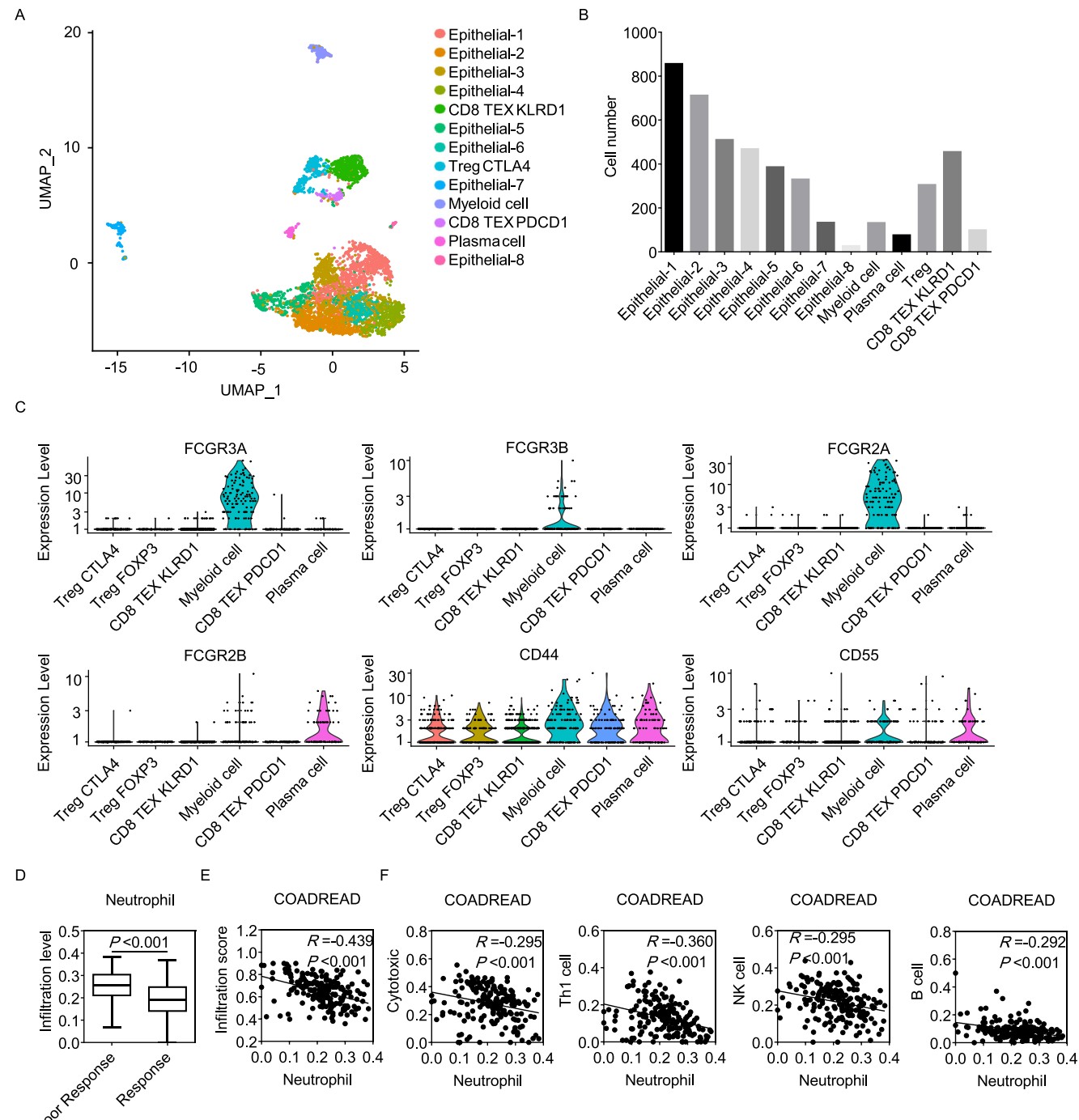

**Fig. 2 | Elevated neutrophil infiltration is associated with inhibited tumor immune status. A**, **B** The 4489 qualified cells were divided into eight epithelial cells and five groups of immune cells using a UMAP plot (**A**). Cell number in each cluster is shown (**B**). **C** The expression level of neutrophil marker genes in each cluster is shown using a violin plot. **D** Data of MSI CRC cases in the TCGA were analyzed using ImmuCellAI. Infiltration levels of neutrophils were compared between patients who predicted poor response ($N = 72$) and good response ($N = 117$) to ICIs. The median

values are shown as the center of the box, and the ranges are displayed by the whiskers. The quartiles are shown as the bounds. The two-tailed unpaired Student's $t$ test was used. **E** Correlation between total infiltration scores and neutrophil infiltration levels was analyzed. The Spearman rank correlation test was used. **F** Correlations between the infiltration levels of cytotoxic T cells, Th1 cells, NK cells, B cells and neutrophils were analyzed. The Spearman rank correlation test was used. Source data are provided as a Source Data file.

resistance to ICIs through neutrophil-associated T-cell exhaustion. In addition, both inflammatory conditions and a high NLR predict a poor response to ICIs, and the prognostic value could be further increased when these two predictors are combined.

Prior to the current study, the influence of neutrophils on immune status and ICI response has been revealed[30]. Tumor-associated neu-trophils are generally considered to promote tumorigenesis among

multiple tumor types, and there are also studies regarding neutrophils establishing a pre-metastatic niche for tumor cells[31–34]. In pancreatic cancer, it has been found that IL17-induced neutrophil extracellular traps could mediate resistance to ICIs[11], and neutrophils in tumors effectively suppress normal T-cell immunity in gastric cancer[35]. In the current study, we have significantly expanded upon these previous observations. The current study demonstrates that both local

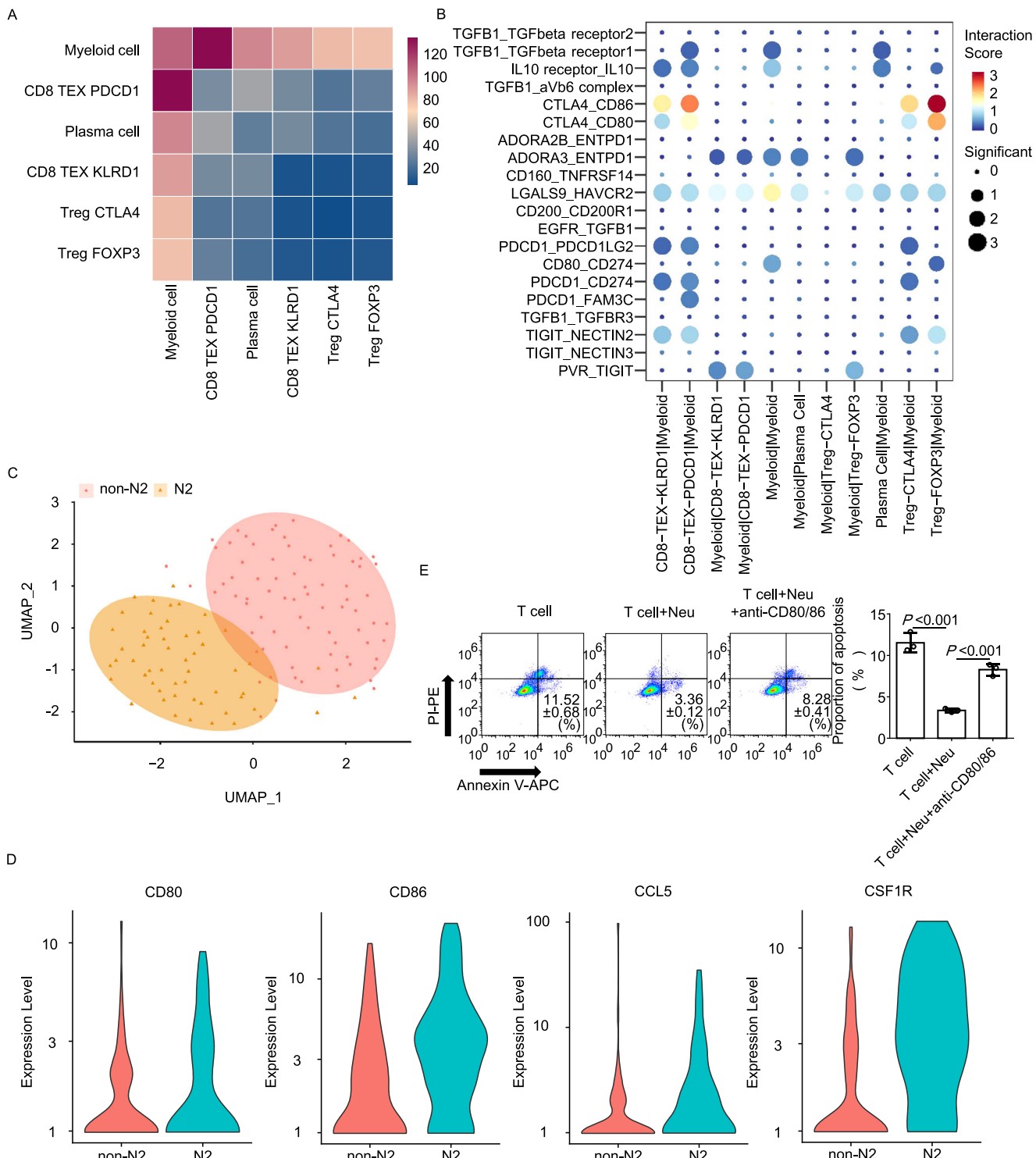

**Fig. 3 | Single-cell sequencing reveals an inhibitory role of neutrophils in tumor immune status through CD80/CD86-CTLA4 axis. A, B** Interactions between subtypes of immune cells were analyzed. Functional phenotypes and predicted interactions of myeloid cells and T cells (**A**), and ligand-receptor interactions in immune checkpoints between immune cells are shown (**B**). **C** Myeloid cells were divided into 2 clusters (N2 and non-N2) using a UMAP plot. **D** The expression level of *CD80, CD86, CCL5* and *CSF1R* in N2 and non-N2 cluster is shown. **E** Murine T cells, T cells pre-stimulated with neutrophils, T cells pre-stimulated with neutrophils in

the presence of CD80 + CD86 neutralizing antibodies (anti-CD80/CD86) were pre-treated with PD-1 neutralizing antibodies and then co-cultured with MC38 cells for 6 h. Apoptosis assays were used to evaluate the proportion of apoptosis in each group. The two-tailed unpaired Student's *t* test was used. The mean values are shown, and the standard deviations are displayed by the error bars. Each experiment was repeated 3 times with 3 replicates. Source data are provided as a Source Data file. Neu neutrophil.

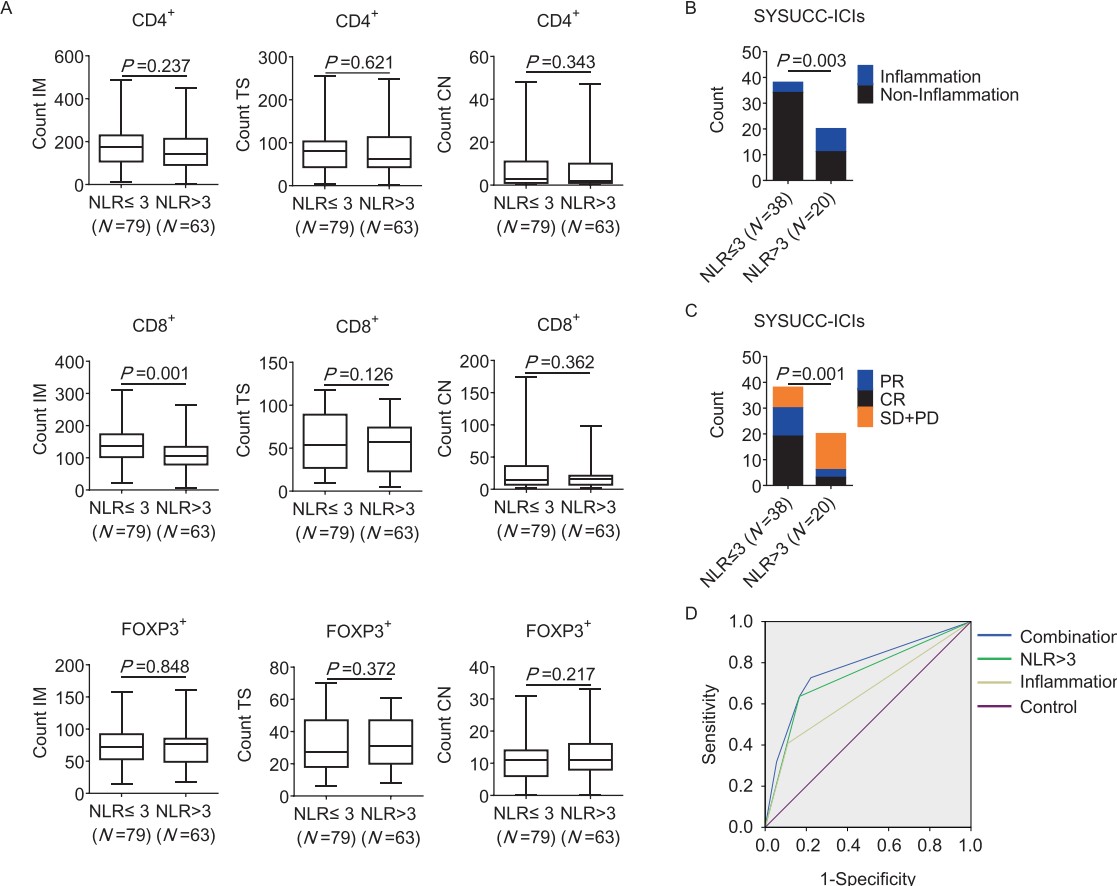

**Fig. 4 | Elevated NLR is associated with poor immune status and resistance to ICIs. A** In a cohort of 142 surgically resected dMMR CRCs, the CD4[+], CD8[+] and FOXP3[+] TILs in the IM, TS and CN were compared between patients with NLR ≤ 3 (*N* = 79) and NLR > 3 (*N* = 63). The two-tailed unpaired Mann-Whitney *U* rank-sum test was used. The median values are shown as the center of the box, and the ranges are displayed by the whiskers. The quartiles are shown as the bounds. **B** In 58 MSI-H CRCs receiving ICIs, the ratio of local inflammation was compared between patients with NLR ≤ 3 (*N* = 38) and *N*LR > 3 (*N* = 20). The Wald chi-square test was used.

**C** Tumor response to ICIs were compared between MSI-H CRC patients with NLR ≤ 3 and NLR > 3. The Wald chi-square test was used. **D** ROC curve analysis was conducted to evaluate the predictive performance of the prognostic features for ICI response. Source data are provided as a Source Data file. IM invasive margin, TS tumor stromal, CN cancer nest, NLR neutrophil-to-lymphocyte ratio, PR partial response, CR complete response, SD stable disease, PD progressed disease, ROC receiver operating characteristic.

inflammatory conditions and NLR > 3 are associated with a poor immune status and a poor tumor response to ICIs, and the cell-cell network analysis in Patient 1 indicates that neutrophils play a prominent role in immunosuppression in MSI-H CRCs.

Although neutrophils can be activated by inflammatory conditions, the current study demonstrates different cytotoxicity between T cells in peripheral blood and TILs, indicating that neutrophils in tumors are functionally distinct from their peripheral counterparts. Apart from our study, Wang et al has also revealed that tumor-infiltrating neutrophils exhibit an activated phenotype compared with normal activated peripheral cohorts[35]. In addition, multiple cell types in the tumor microenvironment could contribute to the pool of cytokines, including G-CSF, GM-CSF, CXCR2 ligands and IL17, which activate neutrophils and educate other immune cells to be tumor-associated[25,35]. Therefore, tumor-infiltrating neutrophils are thought to contribute more to immunosuppression than peripheral neutrophils.

Studies have suggested that the induction of PD-L1 by inflammatory factors may be one of the most important factors affecting the therapeutic efficiency of PD-1 blockade[13,35]. However, the current study indicates that the CTLA4-CD80/CD86 axis could also participate in the interaction of neutrophils and T cells. More significantly, an in vitro experiment demonstrates that neutralization of CD80 and CD86 may reduce the inhibitory function of neutrophils on T cells. Depleting immunosuppressive tumor-associated myeloid cells is an attractive

therapeutic approach to promote antitumor immune responses[36,37]. However, these inhibitors have provided minimal therapeutic benefits in cancer patients as monotherapy, and whether these treatments could restore sensitivity to ICIs remains unclear. Because ICIs directed against PD-1 and CTLA4 are highly effective in dMMR/MSI-H CRCs, and CTLA4 blockade reduces immature myeloid cells in cancers[38], blockade of the CTLA4 axis may be appropriate for MSI-H CRC patients with inflammatory conditions.

CD80/CD86 appears to play a central role in inflammatory diseases[39,40]. As a response to hypoxia and specific cytokines, myeloid cells express elevated levels of CTLA4 ligands and other immune checkpoint inhibitors. CTLA4 ligands such as B7 molecules are also highly expressed in dendritic cells[41]. In T cells, CTLA4 and CD28 exist as homodimers capable of binding to CD80/CD86 via the same extracellular motif. Some researchers have claimed that up-regulating CD80 in neutrophils could promote the activation of T cells by interacting with CD28[42], while others have suggested that CTLA4 has a substantially higher affinity and avidity, thereby outcompeting CD28 and simply preventing it from eliciting its stimulatory signals[38,43]. Together, these findings suggest that the interaction of CTLA4 and CD80/CD86 contributes most to T cell exhaustion under inflammatory conditions.

It has also been shown that elevated neutrophils in peripheral blood predict poor prognosis and resistance to ICIs in multiple cancers[44,45]. In the study of Fan et al, an NLR > 5 was associated with

poor clinical response to anti-PD-1 therapy in patients with advanced gastric and colorectal cancers[46]. In metastatic MSI-H CRC, the pan-immune-inflammation value, calculated using routine blood test data, is a strong predictor of outcome in patients receiving ICIs[47]. In the current study, an NLR > 3 was also identified as a predictor of poor ICI response. The detection of neutrophil frequency, NLR, or neutrophil-releasing factors in patients' serum is easy, inexpensive, and applicable. Additionally, in cancer patients without inflammatory conditions, an elevated NLR is associated with worse immune status and poor ICI response[48]. Since the AUC of the inflammatory condition combined with the NLR was larger than that of single factors, we consider that in MSI-H CRCs, patients with inflammatory conditions could have a poor tumor response to ICIs. Among patients without inflammatory conditions, an NLR > 3 could be a promising predictor for a poor response to ICIs. However, limited sample size in the current study, which could lead to selection bias, restricts the meaning of our findings. The predictive value of these inflammatory features is supposed to be further validated in a larger cohort.

In conclusion, the current study demonstrates that inflammatory conditions in MSI-H CRCs correlate with resistance to ICIs through neutrophil-associated immunosuppression, and additional blockade of CD80/CD86-CTLA4 axis may potentially improve the sensitivity among those patients. Clinically, both inflammatory conditions and an NLR > 3 could predict a poor tumor response to ICIs in MSI-H CRCs, and the value could be further increased when these two predictors are combined.

## Methods

### Patient inclusion and follow-up

Patients diagnosed with MSI-H CRC from Sun Yat-sen University Cancer Center (SYSUCC, Guangzhou, China), who started to receive PD-1 blockade therapy from January 2017 to October 2020 were retrospectively enrolled. The inclusion criteria were as follows: (1) pathologically diagnosed colorectal cancer; (2) genetically diagnosed MSI-H; (3) receiving PD-1 blockade. The exclusion criteria were as follows: (1) receiving only postoperative PD-1 blockade after radical surgery; (2) receiving less than 2 courses of treatment; (3) without determination of tumor response. Sixty-eight patients were included, while 4 patients were excluded for receiving only postoperative treatment, 1 for receiving 1 course of treatment and 1 for without determination of tumor response in our center. Finally, 62 patients were enrolled. In another cohort of 142 dMMR/MSI-H CRCs, patients who received surgical treatment with preoperative blood test data and sufficient tumors for counting TILs were enrolled as previously reported[28]. Follow-up data, blood test, CT scanning, genetic diagnosis, pathology report, and determination of responses to ICIs were collected from the tracking system. Last follow-up date was June 14, 2022. The end point of PFS was defined as tumor progression or the last follow-up. For patients receiving ICIs, those with consistent peripheral NLR > 3 in at least 2 courses of treatment were considered to have NLR > 3. Clinical tumor responses were determined as CR, PR, SD and PD by radiologists. Pathological tumor responses were determined using TRG grading scale. TRG 0 is defined as no residual tumor cells found microscopically on multiple consecutive sections. TRG 1 is defined as the presence of only small clusters of tumor cells that can be observed under the plasma membrane. TRG 2 is defined as fibrosis within the tumor lesion and the observation of fragmented residual tumor cells. TRG 3 was defined as a lesion with little to no fibrosis and no change in the number of tumor cells. Patients who had obstructions, perforations, peritonitis, or other radiologically diagnosed inflammation in abdominal viscera or metastatic sites were considered to have local inflammatory conditions. All procedures performed in studies involving human participants and human material have been approved by the ethical standards of the Ethics Committee of Sun Yat-sen University Cancer Center (GZR2020-273) and were in accordance with the 1964 Helsinki declaration and its later amendments or comparable ethical standards. Informed consent was obtained from all individual participants included in the study. Reporting of clinical data complies with the STROBE guidelines.

### Organoids

Tumor tissues of 4 MSI-H CRCs were collected. Tissues from Patient 1, 2, 3 and 4 were collected after PD-1 blockade. Tumors were digested with digestion buffer (RPMI 1640 medium [Gibco] containing 10% fetal bovine serum [FBS], 1% penicillin-streptomycin, 4 mg/mL collagenase [Sigma C5138]) and embedded in Matrigel (Corning). After solidification, the Matrigel was overlaid with IntestiCult OGM Human (Stem Cell) supplemented with penicillin (100 U/mL), streptomycin (100 μg/mL) and 10 mM Y-27632 (Sigma-Aldrich) at 37 °C with 5% $CO_2$. Organoids used in experiments were under passage 30. For passage, culture medium was removed, and organoids were digested into single cells using TrypLE Express (Gibco). After removal of TrypLE Express and digested Matrigel, Organoid cells were embedded in fresh Matrigel with certain concentration. For cryopreservation, Organoid were digested into single cells and cryopreserved at -80 °C in FBS containing 10% dimethyl sulfoxide (DMSO). Hematoxylin and eosin (H&E)-stained sections of organoids were assessed by pathologists to determine the tumor status. For H&E staining, Matrigel samples were fixed with formalin at 4 °C overnight and coated with a 5% agarose gel before paraffin imbedding.

### Preparation and treatment of human T cells

Peripheral blood mononuclear cell (PBMC)-derived T cells from 4 MSI-H CRC patients were isolated, and TILs from surgical-resected tissues were obtained after PD-1 blockade, using a human pan-T cell isolation kit according to the manufacturer's instructions (Miltenyi Biotec). Human T cells were cultured in Human ImmunoCult-XF T Cell Expansion medium (Stem Cell) with penicillin (100 U/ml) and streptomycin (100 μg/ml) at 37 °C with 5% $CO_2$. Cells were prestimulated with IL-2 (200 U/ml, Peprotech), anti-CD3 (5 μg/mL, Peprotech) and anti-CD28 (5 μg/mL, Peprotech) for 48 h in the presence of paired CRC organoid cells at a 20:1 ratio.

### Organoid-T cell co-culture

To evaluate the cytotoxicity of T cells, organoids were dissociated into single cells and plated ($1 \times 10^5$ per well) in a 24-well plate in the absence of Matrigel 24 h before co-culture. Pretreated T cells ($2 \times 10^6$) were added to each plate. After 6 h, tumor cells were obtained. Using an Annexin V Apoptosis Detection Kit (Dojindo), the cells were stained according to the manufacturer's instructions, and then resuspended in phosphate buffered saline (PBS). Cell apoptosis analysis was conducted using a Beckman CytoFLEX FCM (Beckman Coulter) with the software CytExpert 2.0 (Beckman Coulter). The proportion of apoptosis was calculated using FlowJo V10 (BD). Each experiment contained 3 replicates.

### Preparation and treatment of murine neutrophils and lymphocytes

Neutrophils from female C57BL/6 mice (8 weeks old) were derived from the bone marrow using the mouse Neutrophil Isolation Kit (Miltenyi Biotec), and were cultured in RPMI 1640 supplemented with 10% FBS, penicillin (100 U/ml), and streptomycin (100 μg/ml) at 37 °C with 5% CO2. Neutrophils were stimulated with mouse recombinant GM-CSF (100 ng/mL, Peprotech) for 24 h, in the presence or absence of anti-CD80 (2 μg/mL, R&D System) and anti-CD86 neutralizing antibody (2 μg/mL, R&D System).

T cells for MC38-T cell co-culture were isolated from the spleens of mice with mouse pan-T cell isolation kits (Miltenyi Biotec) according to the manufacturer's instructions. T cells were cultured in RPMI 1640 supplemented with 10% FBS, penicillin (100 U/ml), and streptomycin (100 μg/ml) at 37 °C with 5% CO2. Before experiments, T cells were

prestimulated with mouse IL-2 (200 U/ml, Peprotech), anti-CD3 (1 μg/mL, Peprotech) and anti-CD28 (1 μg/mL, Peprotech) in the presence of MC38 cells at a 20:1 ratio for 48 h. T cells were also treated with anti-PD-1 neutralizing antibody (1 μg/ml, BioXcell) for 24 h. For neutrophil-T cell co-culture, T cells were co-cultured with pre-stimulated neutrophils at a 2:1 ratio in RPMI-1640 medium for another 24 h[35].

## MC38-T cell co-culture

MC38 cell is a gift from Dr. Xueyi Zheng. MC38 cells were cultured in complete medium (RPMI 1640 [Gibco] supplemented with 100 μg/mL streptomycin, 100 IU/mL penicillin, and 10% FBS) at 37 °C in 5% CO2. Before experiment, tumor cells were plated in a 24-well plate and left overnight in the presence of 200 ng/ml mouse IFN-γ (Peprotech). T cells were added to the tumor cells at a 20:1 ratio at 37 °C and co-cultured for 6 h. Using an Annexin V Apoptosis Detection Kit (Dojindo), the MC38 cells were stained according to the manufacturer's instructions and resuspended in PBS for cell apoptosis analysis using a Beckman CytoFLEX FCM with the software CytExpert 2.0. The proportion of apoptosis was calculated using FlowJo V10. Each experiment contained 3 replicates.

## scRNA-seq

The scRNA-seq was conducted using the primary tumor of Patient 1, 2, 3 and 4. An scRNA-seq library was prepared using the DNBelab C4 system[49]. Briefly, the single-cell suspension was transformed into the scRNA-seq library of barcodes through the steps of droplet encapsulation, emulsification and fragmentation, mRNA capture bead collection, reverse transcription, cDNA amplification and purification. An indexed sequencing library was constructed according to the manufacturer's instructions. The sequencing library was quantified using the Qubit SSDNA Assay Kit (Thermo). DIPSEQ T1 was used for sequencing libraries at the National Gene Bank (CNGB, BGI-SHENZHEN, Shenzhen, China). The read structure was paired with Read 1 and Read 2. Read 1 contained 30 bases, including 10 base pair (bp) cell barcode 1, 10 bp cell barcode 2 and 10 bp unique molecular identifier (UMI), and Read 2 contained 100 transcriptional base sequences and a 10 bp sample index. The raw FASTQ files were processed by DNBelab_C_Series_HT_scRNA-analysis-software (https://github.com/MGI-tech-bioinformatics/DNBelab_C_Series_HT_scRNA-analysis-software)[50]. Briefly, the FASTQ raw data were converted to a Cell Ranger-specific FASTQ file, then the converted FASTQ files were aligned to GRCH38 human reference using STAR software (v2.5.3)[51]. The mapped reads were then filtered out for valid cell barcodes and UMIs to generate a gene-cell matrix for downstream analysis.

## Unsupervised clustering and cell type annotation

Cell clustering was conducted by the Seurat (v3.1)[52] package in RStudio (v1.1.383). Genes expressed in less than 3 cells were filtered out, and cells with fewer than 500 or more than 10,000 genes were excluded. The 3 libraries were then integrated using the "Merge" functions, and the batch effects were checked if the cells were separately distributed with the "DimPlot" function. Then, the integrated data were scaled to calculate the principal component analysis. The first 30 PCs were used to construct the SNN network, and the graph-based clustering method Louvain algorithm was used to identify the cell clusters with a resolution of 0.6. Finally, UMAP was used to visualize the clustering results in two-dimensional space. To annotate each cluster as a specific cell type, we selected some classic markers of immune cells, epithelial cells and fibroblasts. The cell types were annotated using a violin diagram.

## Cell-cell interaction analysis

To analyze cell-to-cell interactions, we used CellPhoneDB[53] to identify significant ligand-receptor pairs in samples from Patient 1 to 4. For immune cells, L-R-specific interactions between cell types were determined based on the specific expression of the receptors of one cell type and the ligands of another cell type. The interaction score refers to the total average of the average expression value of each ligand-receptor pair in the corresponding cell type interaction pair. The expression amount of any complex exported by CellPhoneDB[53] was calculated as the sum of the expression of the constituent genes.

## Analysis of online data

Data on MSI status in ImmuCellAI were obtained from TCGA. The predicted ICI response, infiltration levels of neutrophils, cytotoxic cells, T helper 1 (Th1) cells, natural killer (NK) cells and B cells of MSI CRC cases were also obtained from ImmuCellAI (http://bioinfo.life.hust.edu.cn/ImmuCellAI/#!/).

## IHC analysis and lymphocyte counting

IHC staining for lymphocyte counting was previously conducted by staining for CD4 (with 1:50 dilution, clone 1F6, ORIGENE), CD8 (with 1:80 dilution, clone SP16, DRM012, ORIGENE) and FOXP3 (with 1:100 dilution, clone 236 A/E7, ab20034, Abcam)[28]. Staining for neutrophil counting was conducted using anti-CD11b (with 1:4000 dilution, clone EPR1344, ab133357, Abcam). All specimens were prepared as 4 μm FFPE sections. The sections were deparaffinized via a series of decreasing concentrations of ethanol, deionized with H2O, and rinsed in PBS. Endogenous peroxidase activity was blocked via incubation in 3% H2O2 solution in methanol. The antigenic epitopes were unmasked in a decloaking chamber using citrate buffer (10 mM sodium citrate and 0.05% Tween 20, pH 6). The sections were then washed in deionized water, rinsed in PBS, blocked for 30 min at room temperature with 5% bovine serum albumin in PBS, and incubated with primary antibodies in a humidified chamber at 4 °C overnight. After washing, the sections were incubated with anti-rabbit/mouse IgG monoclonal antibody (DAKO Real Envision) at room temperature for 1 h. Staining was performed using DAB (DAKO Real Envision), followed by counterstaining using hematoxylin. The number of lymphocytes was counted by a pathologist according to the following method:[28] select five high-power fields (HPF) in the invasive margin (IM), tumor stromal (TS), and cancer nest (CN); count the positive cells; and take the average (Olympus BX41). The number of neutrophil in HPF was counted by a pathologist according to the following method: select five HPFs in the tumor area (for patient with residual tumor) or previous tumor site (for patient with complete response); count the positive cells; and take the average (Olympus IX73).

## Statistical analysis

SPSS 19.0 (Chicago, IL) and GraphPad Prism 6 (San Diego, CA) were used for data analysis. Data for continuous and discrete variables are reported as the mean and median respectively. Data for categorized variables are reported as percentages. Shapiro-Wilk test was used for test of Gaussian distribution. Student's $t$ test was used for the comparison of two sets of quantitative data that subject to the Gaussian distribution. The Mann-Whitney $U$ rank-sum test was used for the comparison of two sets of quantitative data that deviate from the Gaussian distribution. For Student's $t$ test, the mean value is shown, and the standard deviation ($SD$) is displayed by the error bar (mean ± $SD$). For the Mann-Whitney $U$ rank-sum test, the median value is shown, and the range is displayed by the error bar. The Wald chi-square test was used to compare the differences in categorical parameters. Distributions of PFS were determined by Log-rank test using Kaplan-Meier methods. Univariate and multivariable Cox proportional hazards models were used to predict the outcomes of influential factors. The Spearman correlation test was used to measure the relationship between two variables that do not deviate from the Gaussian distribution. All hypothesis tests were two-sided, and those with $P$ value <0.05 were considered statistically significant. ROC curves were constructed to quantify the predictive performance of the prognostic factors for ICI response by assessing the respective AUC with the 95% $CI$.

## Reporting summary

Further information on research design is available in the Nature Portfolio Reporting Summary linked to this article.

## Data availability

The raw scRNA-seq data in this study have been deposited at the Gene Expression Omnibus (GEO) database under accession code GSE179784. The GRCH38 human reference are available at UCSC Genome Browser Home [http://genome.ucsc.edu/]. The referenced raw data from TCGA are available at TCGA database [https://portal.gdc.cancer.gov/], and processed data are available within the Source data. The clinical data and experiment data in this study are provided in the Source Data file. A reporting summary for this article is available as a Supplementary Information file. Source data are provided with this paper.

## Code availability

The software and R package used in this paper are public or previously reported tools. The methods refer to the official suggestions of the tools, and the pipeline running the tools were described in the method section of the manuscript, which did not involve the newly developed codes or tools.

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

## Acknowledgements

We thank Dr. Jianhui Yue from BGI-SHENZHEN (Shenzhen, China) for his assistance with data analysis. We thank Dr. Xueyi Zheng from Sun Yat-sen University Cancer Center for giving us MC38 cells. We thank the staff at American Journal Experts (North Carolina, USA) for their assistance with English writing. This study was funded by the National Natural Science Foundation of China (grant numbers 82102873 by J.T., grant numbers 82073159 and 81871971 by PR.D.) and Guangdong Basic and Applied Basic Research Foundation (grant numbers 2020A1515110544 by J.T).

## Author contributions

Conceptualisation: Q.S., X.Z. and PR.D.. Methodology: Q.S., X.Z., C.C., J.Y., W.L., K.H., W.J., L.L., L.K., Y.L., Z.H., C. Zhou., C. Zhang., L.Z., B.X., W.M., Y.X. and J.Q.. Data curation: X.Z., C.C., K.H., W.J., L.L., B.X., J.Z. and Z.P.. Writing—original draft preparation: Q.S. and X.Z.. Writing—review and editing: J.T., K.H. and PR.D.. Supervision: X.Z., Z.P. and PR.D.. Project administration: X.Z., Z.P. and PR.D.. Funding acquisition: J.T. and PR.D.. Q.S., X.Z., C.C., J.T., J.Y. and W.L. are joint first authors.

## Competing interests

The authors declare no competing interests.

## Additional information

[1]Department of Colorectal Surgery, Sun Yat-sen University Cancer Center, Guangzhou 510060, P. R. China. [2]State Key Laboratory of Oncology in South China, Guangzhou 510060, P. R. China. [3]Collaborative Innovation Center for Cancer Medicine, Guangzhou 510060, P. R. China. [4]BGI-Shenzhen, Shenzhen 518120, P. R. China. [5]Department of Thoracic Surgery, Peking University Shenzhen Hospital; Shenzhen Peking University-The Hong Kong University of Science and Technology Medical Center, Shenzhen 518035, P. R. China. [6]Department of Experimental Research, Sun Yat-sen University Cancer Center, Guangzhou 510060, P. R. China. [7]These authors contributed equally: Qiaoqi Sui, Xi Zhang, Chao Chen, Jinghua Tang, Jiehai Yu, Weihao Li ✉e-mail: dingpr@sysucc.org.cn

