## [Peer Review File · Nature Communications]

Inflammations promote resistance to immune checkpoint inhibitors in high microsatellite instability colorectal cancerREVIEWER COMMENTS

Reviewer #1 (Remarks to the Author):

The study from Sui et al. investigates the impact of inflammatory condition on the response to immune checkpoint inhibitors (ICI) in dMMR/MSI-H colorectal cancers (CRC). They firstly showed a case (patient #1) of dMMR/MSI-H CRC with local inflammation. The patient responded poor to ICI at primary tumor but responded good at metastatic liver tumors. A clinical cohort analysis showed that inflammatory conditions during ICI treatment were correlated with a worse response in dMMR/MSI-H CRC. Co-culture of patient-derived organoids and TIL/PBMC showed that in the patient with inflammatory condition, the apoptotic proportion of TIL was reduced without affecting the PBMC apoptosis. Single cell RNA-seq of the tumor sample from patient #1 showed that myeloid cells had a core role in microenvironments with intense interaction with other types of immune cells. They identified that CTLA4-CD80/86 was a prominent receptor-ligand interaction and it was selected for further study. Additional treatment of α -CTLA4 with α -PD1 augmented the apoptotic proportion of murine CRC CT26 cells co-cultured with T cells. Next, they analyzed the TCGA database and showed that elevated neutrophil infiltration was associated with a poorer response to ICI in MSI-H CRC. Finally, they examined the neutrophil-lymphocyte ratio (NLR) in 142 resected dMMR/MSI-H CRC patients. A significant lower TILs was noted in patients with $NLR > 3$. In 73 patients receiving ICI treatment, combination of the inflammatory conditions and NLR can be a marker for predicting the ICI response.

This is an interesting study which asked an important clinical question and mainly used clinical materials to support the hypothesis. However, the major weakness of the study is that the authors only provided the observation data without proving their concept in vivo. Furthermore, the linkage between "inflammatory conditions", which means systemic inflammation presented as a higher NLR, and increased infiltration of neutrophils in tumors is weak. The phenotype and functions of infiltrated neutrophils are not characterized. The specific comments are listed below.

1. Table S1 is the characteristics of "73 MSI-H CRC". However, there is a significant proportion of MSI-L/MSS (4 cases with inflammatory conditions and 8 cases without inflammatory conditions) and undetermined dMMR (totally 7 cases). If the study focuses on the dMMR/MSI-H CRC, these patients should be removed and the data should be reanalyzed. Furthermore, why the MSI-L/MSS cases were treated with ICI?

2. For the receptor-ligand interaction analysis, CTLA4-CD80/86 is not the most significant one in the single cell RNA-seq analysis. The rationale for selecting this interaction without considering other ones should be explained.

3. In Figure 1E, the authors suggested that "local inflammation without systemic immunosuppression is noted in the inflammatory conditions of CRC patients". However, in Figure 4 they showed that a higher NLR is associated with a lower CD4+ and CD8+ TILs. The relationship between the systemic inflammation, local TILs, and response to ICI should be clarified.

4. The phenotype and activities of tumor-infiltrated neutrophils (TINs) are not examined in this study. NET has been noted to be associated with local immunosuppression in tumors. The N2 neutrophil phenotype is noted to be associated with immunosuppression of tumor microenvironments. The author may need to characterize the TINs in this study.

5. The study lacks of the in vivo validation, which will be crucial for supporting the hypothesis. For example, observation of the impact of depletion of neutrophils on ICI response in a syngeneic dMMR/MSI-H model will be important.

Reviewer #2 (Remarks to the Author):

Peer review:

Inflammatory Conditions Promote Resistance to Immune check point Inhibitors in High Microsatellite Instability Colorectal Cancer

Summary:

Sui et al. investigate the impact of having an inflammatory condition on the prospects of MSI-H CRC. The paper describes a case story involving an MSI-H CRC patient with primary tumor

progression, but liver metastasis regression on pembrolizumab treatment (Fig 1A). The authors investigate the impact of inflammatory condition on ICI response rates and find a significantly higher number of PR and CR in patients with no inflammatory conditions in a cohort of 73 dMMR/MSI-H CRC patients (Fig 1B-C). By setting up organoid and T-cell co-culture for the case study patient, they show that PBMCs over TILs show reactivity towards the autologous organoids (Fig 1D-E). This phenomenon is reversed in another patient showing complete response on ICI and no inflammatory condition.

scRNA sequencing analysis of the case-patient points to neutrophil-like myeloid cells as a central player in cell-cell interactions within this tumor and to CTLA4-CD86 as a central pathway for myeloid-CD8 T cell and myeloid-Treg interactions (Fig. 2A-E).

In a murine system of BALB/c derived immune cells and CT26 cancer cells, they show that neutrophils dampen T-cell mediated killing of CT26 and that this can be reverted by adding anti-CTLA4. From data on MSI CRC in TCGA, the authors show that high neutrophil infiltration correlates with a poor response to ICI, low infiltration of other immune cells and poor overall survival (Fig 3A-E).

For the final main figure (Figure 4), the authors use a cohort of 142 resected dMMR CRCs to investigate the correlation between neutrophil-lymphocyte ratio (NLR) in peripheral blood with T-cell tumor infiltration. Here, they find that patients with a $NLR > 3$ has inferior CD8+ T-cell infiltration in the invasive margin (IM) and the cancer nest (CN). In the ICI-treated cohort from Figure 1, they see an increased NLR in patients with an inflammatory condition and a decreased response rate as well as inferior progression-free survival in patients with $NLR > 3$. In conclusion, the authors suggest that inflammatory conditions correlate with resistance to ICIs through neutrophil-associated immunosuppression and suggest anti-CTLA4 to be added to their treatment. Furthermore, they suggest to include inflammatory conditions and $NLR > 3$ as predictive biomarkers for ICI treatment of MSI-H CRCs.

Reviewer comments:

What are the noteworthy results?

The authors show that MSI-H CRC patients with an inflammatory condition is a subgroup of patients with a significantly worse response to ICI. Through a case study, they identify neutrophils as an important mediator for immune suppression.

Will the work be of significance to the field and related fields? How does it compare to the established literature? If the work is not original, please provide relevant references.

The work is in line with the established literature and builds on already published results concerning NLR and the combinatorial benefit of CTLA-4 and PD-L1 inhibition for PD-L1 therapy resistant patients.

The claim that inflammatory conditions are the cause of resistance to MSI-H CRC ICI resistance is new and of interest to the field.

Does the work support the conclusions and claims, or is additional evidence needed?

For their primary claim the authors use a cohort of 73 patients treated with ICI. The patients are primarily MSI, but approx. 30 % are either MSS or undetermined dMMR status, which may influence the data. In addition, 50 % received ICI monotherapy and 50 % did not. The authors should state more clearly which treatment the patients received instead (table 1). It would be of great benefit to the manuscript if the authors could validate their findings in another ICI treated cohort.

In figure 1D-E, the authors use organoids from three patients with different response to treatment eg. PD, SD and CR. To fully support their claim about local immune suppression within tumors from patients with inflammatory conditions and differential ability to activate T-cells mimicking treatment response, the authors need to include more patient-organoid and immune cells for each condition.

The findings from scRNA are based on one patient, and there is no data from patients without an inflammatory condition to compare with. To fully support their claim, the authors should include

more patients with inflammatory conditions as well as patients without inflammatory conditions in the scRNA analysis.

Are there any flaws in the data analysis, interpretation and conclusions? - Do these prohibit publication or require revision?

In figure 2F the authors investigate the effect of a-CTLA4 and neutrophils on T-cell mediated cancer cell killing. They observe an increased apoptosis rate of tumor cells upon addition of CTLA4. Preincubation of T-cells with neutrophils gives a slight decrease in tumor-killing, which is abrogated by addition of CTLA4. The authors claim that this supports the hypothesis of neutrophils inhibiting the T-cells through CD86-CTLA4 interaction. The change in apoptosis is, however, also present without neutrophils, indicating that it is a matter of an a-CTLA4 effect on T-cells independent of neutrophils.

Since the hypothesis of neutrophils inducing immune suppression through CD86-CTLA4 interaction is based on scRNA analysis from only one tumor and the link between neutrophil-CD86 and CTLA-4 induced inhibition is not clearly made, the conclusion needs more data to be substantiated.

In the conclusion (L235) the authors suggest adding a-CTLA4 for a better ICI sensitivity in MSI-H CRC patients with inflammatory conditions, but since approx. 50 % of the 73-patient cohort did not receive monotherapy and may hence have already received combination therapy together with the unknown treatment regime for the TCGA cohort, the authors should clarify this point and consider revising the conclusion if not in line.

Is the methodology sound? Does the work meet the expected standards in your field?

The methods used in the study are sound. The authors include both clinical as well as translational data to support their study.

The in vitro data using organoids is, however, based on few/single samples which makes the conclusions too vulnerable, which also goes for the scRNA analysis. To meet the standards of the field more samples will need to be included.

For the organoid cultures, the authors should include how the organoids were validated eg. did they do SNP analysis or similar to check the match with the donor patient. Were the organoids cryopreserved and how?

Figure 1E: the authors should include controls of organoids alone to see the background apoptotic signal for each organoid line.

Is there enough detail provided in the methods for the work to be reproduced?

The method section is clearly written. For the work to be reproduced, it would be beneficial if the number of cells used for in vitro experiments was stated. Also, the concentration of some reagents fx a-CD3 and a-CD28 (L294) are missing and some manufacturers fx Annexin V Apoptosis Detection Kit (L302). For the flow cytometry analysis, the authors should include their gating strategy, preferably as a supplementary figure. The statistics section is comprehensive, but it would be beneficial if the statistical methods used was stated in the figure legends.

Further comments:

Abstract: the abstract would benefit from a clearer statement of the specific research question for this paper.

L59: In the introduction the authors state that inflammatory conditions are common complications in CRCs. It would be nice if the authors would support this with a reference.

L111: the authors mention a supplementary table. I have not been able to find this among the documents.

L87: the MSI-H cohort. The authors describe the cohort as being an MSI-H cohort albeit 30 % of the patients are not MSI-H. This may be a bit confusing. The authors should consider rephrasing.

L140: the TCGA cohort. If possible, the authors should include information about the ICI treatments used for this cohort.

Figure 1A: please use arrows to highlight the tumor/metastases

Overall assessment:

The research question is exciting and interesting and the methodology in the manuscript is just and embarks on highly relevant techniques. I do however believe that more data is needed to draw the conclusions of the manuscript and support the claims before publication. I would encourage the authors too include some or all of the suggestions as stated about in order to do so.

Reviewer #3 (Remarks to the Author):

The authors explore the relation between inflammatory conditions in MSI-H CRCs and resistance to ICIs through neutrophil-associated immunosuppression. I will focus on some aspects of the paper.

On Line 88: " We found that inflammatory conditions during ICI treatment were correlated with a higher ratio of stable disease (SD) and PD (87.5% vs. 33.3%, $P < 0.001$) and worse progression-free survival (PFS) ($P = 0.001$) (Figure 1B and C)."

It is unclear the test performed and how. Please explain.

On Line 385: "Student's t test was used for the comparison of two sets of quantitative data that deviated from the Gaussian distribution. The Mann-Whitney U rank-sum test was used for the comparison of two sets of quantitative data that did not deviate from the Gaussian distribution. " This statement seems incorrect, I guess you switched the non deviated with deviated. How did you measure the deviation? Please explain.

On line 327: "cells with fewer than 500 or more 328 than 10,000 genes were excluded". Can you explain these upper and lower bound? Particularly relevant is why you excluded the cells with more than 10.000 expressed genes.

On line 396: "The Pearson or Spearman rank correlation test was used to measure the relationship between two variables" please specify which did you use and when.

On line 397: " All P values were two-sided" This sentence is incorrect. Hypothesis tests can be two sided, not the p-values.

I would like to have more details on the data pre-processing and on the clustering approaches. A comparison of at least two approaches would be nice.

Author response to REVIEWER COMMENTS

Reviewer #1 (Remarks to the Author):

The study from Sui et al. investigates the impact of inflammatory condition on the response to immune checkpoint inhibitors (ICI) in dMMR/MSI-H colorectal cancers (CRC). They firstly showed a case (patient #1) of dMMR/MSI-H CRC with local inflammation. The patient responded poor to ICI at primary tumor but responded good at metastatic liver tumors. A clinical cohort analysis showed that inflammatory conditions during ICI treatment were correlated with a worse response in dMMR/MSI-H CRC. Co-culture of patient-derived organoids and TIL/PBMC showed that in the patient with inflammatory condition, the apoptotic proportion of TIL was reduced without affecting the PBMC apoptosis. Single cell RNA-seq of the tumor sample from patient #1 showed that myeloid cells had a core role in microenvironments with intense interaction with other types of immune cells. They identified that CTLA4-CD80/86 was a prominent receptor-ligand interaction and it was selected for further study. Additional treatment of α -CTLA4 with α -PD1 augmented the apoptotic proportion of murine CRC CT26 cells co-cultured with T cells. Next, they analyzed the TCGA database and showed that elevated neutrophil infiltration was associated with a poorer response to ICI in MSI-H CRC. Finally, they examined the neutrophil-lymphocyte ratio (NLR) in 142 resected dMMR/MSI-H CRC patients. A significant lower TILs was noted in patients with $NLR > 3$. In 73 patients receiving ICI treatment, combination of the inflammatory conditions and NLR can be a marker for predicting the ICI response.

This is an interesting study which asked an important clinical question and mainly used clinical materials to support the hypothesis. However, the major weakness of the study is that the authors only provided the observation data without proving their concept in vivo. Furthermore, the linkage between “inflammatory conditions”, which means systemic inflammation presented as a higher NLR, and increased infiltration of neutrophils in tumors is weak. The phenotype and functions of infiltrated neutrophils are not characterized. The specific comments are listed below.

1. Table S1 is the characteristics of “73 MSI-H CRC”. However, there is a significant proportion of MSI-L/MSS (4 cases with inflammatory conditions and 8 cases without inflammatory conditions) and undetermined dMMR (totally 7 cases). If the study focuses on the dMMR/MSI-H CRC, these patients should be removed and the data should be reanalyzed. Furthermore, why the MSI-L/MSS cases were treated with ICI?
Response: Thank you for your advice. We have removed the MSI-L/MSS and undetermined dMMR cases, and we have also included new MSI-H cases. The latest cohort includes 62 MSI-H CRC patients (Line 87-89, Table 1). Among MSI-L/MSS cases, some of those patients had TMB-H, thus receiving ICI.

2. For the receptor-ligand interaction analysis, CTLA4-CD80/86 is not the most significant one in the single cell RNA-seq analysis. The rationale for selecting this interaction without considering other ones should be explained.

Response: Thank you for your constructive advice. We have mentioned the reason in the manuscript (Line 173-176). Due to that CD80/CD86-CTLA4 axis were involved in the crosstalk between myeloid cells and T cells, and that the application of CTLA4 blockade has been widespread in tumor immunotherapy, we investigated the value of targeting CD80/CD86-CTLA4 axis under inflammatory conditions.

3. In Figure 1E, the authors suggested that “local inflammation without systemic immunosuppression is noted in the inflammatory conditions of CRC patients”. However, in Figure 4 they showed that a higher NLR is associated with a lower CD4+ and CD8+ TILs. The relationship between the systemic inflammation, local TILs, and response to ICI should be clarified.

Response: Thank you for your suggestion. We have revised the writing. The current study demonstrates that neutrophil infiltration could inhibit immune status and result in poor response to ICIs, by analyzing the scRNA seq data of a patient with local inflammation. Both local and systemic inflammation could result in elevated NLR, which is associated with elevated neutrophil infiltration in tumors according to existing studies (He, W. Z. et al. 2018). Besides, in the latest version of the manuscript, single-cell RNA sequencing revealed the infiltration of myeloid cells (including neutrophils) in the tumor of Patient 3 (Supplementary Figure 5A), who had no inflammatory condition with $NLR > 3$ (Line 186-190). She was diagnosed SD after receiving ICI (Supplementary Figure 5A), which further support our point. Apart from that, ROC curves suggests that both inflammatory conditions and an $NLR > 3$ could predict a poor tumor response to ICIs in MSI-H CRCs, and the AUC was further increased when these two predictors were combined (Figure 4D). Therefore, it is significant to reveal the correlation between NLR, immune status and response to ICIs.

4. The phenotype and activities of tumor-infiltrated neutrophils (TINs) are not examined in this study. NET has been noted to be associated with local immunosuppression in tumors. The N2 neutrophil phenotype is noted to be associated with immunosuppression of tumor microenvironments. The author may need to characterize the TINs in this study.

Response: Thank you for your constructive advice. We have further clustered the myeloid cells. The analysis demonstrates that expression level of *CD80*, *CD86*, *CCL5*, and *CSF1R* was much higher in cells of N2 cluster, which indicated that N2-neutrophils participate more in the immune suppression (Figure 3C, D and Supplementary Figure 8).

5. The study lacks of the in vivo validation, which will be crucial for supporting the hypothesis. For example, observation of the impact of depletion of neutrophils on ICI response in a syngeneic dMMR/MSI-H model will be important.

Response: Thank you for your constructive advice. However, in vivo validation following constructing syngeneic dMMR/MSI-H model could cost more than 6

months, which we cannot afford. Instead, we have included more organoid experiments and scRNA-seq (Figure 1 D-F, Supplementary Figure 4-7).

Reviewer #2 (Remarks to the Author):

Peer review:

Inflammatory Conditions Promote Resistance to Immune check point Inhibitors in High Microsatellite Instability Colorectal Cancer

Summary:

Sui et al. investigate the impact of having an inflammatory condition on the prospects of MSI-H CRC. The paper describes a case story involving an MSI-H CRC patient with primary tumor progression, but liver metastasis regression on pembrolizumab treatment (Fig 1A). The authors investigate the impact of inflammatory condition on ICI response rates and find a significantly higher number of PR and CR in patients with no inflammatory conditions in a cohort of 73 dMMR/MSI-H CRC patients (Fig 1B-C). By setting up organoid and T-cell co-culture for the case study patient, they show that PBMCs over TILs show reactivity towards the autologous organoids (Fig 1D-E). This phenomenon is reversed in another patient showing complete response on ICI and no inflammatory condition.

scRNA sequencing analysis of the case-patient points to neutrophil-like myeloid cells as a central player in cell-cell interactions within this tumor and to CTLA4-CD86 as a central pathway for myeloid-CD8 T cell and myeloid-Treg interactions (Fig. 2A-E).

In a murine system of BALB/c derived immune cells and CT26 cancer cells, they show that neutrophils dampen T-cell mediated killing of CT26 and that this can be reverted by adding anti-CTLA4. From data on MSI CRC in TCGA, the authors show that high neutrophil infiltration correlates with a poor response to ICI, low infiltration of other immune cells and poor overall survival (Fig 3A-E).

For the final main figure (Figure 4), the authors use a cohort of 142 resected dMMR CRCs to investigate the correlation between neutrophil-lymphocyte ratio (NLR) in peripheral blood with T-cell tumor infiltration. Here, they find that patients with a NLR>3 has inferior CD8+ T-cell infiltration in the invasive margin (IM) and the cancer nest (CN). In the ICI-treated cohort from Figure 1, they see an increased NLR in patients with an inflammatory condition and a decreased response rate as well as inferior progression-free survival in patients with NLR>3. In conclusion, the authors suggest that inflammatory conditions correlate with resistance to ICIs through neutrophil-associated immunosuppression and suggest anti-CTLA4 to be added to their treatment. Furthermore, they suggest to include inflammatory conditions and NLR>3 as predictive biomarkers for ICI treatment of MSI-H CRCs.

Reviewer comments:

What are the noteworthy results?

The authors show that MSI-H CRC patients with an inflammatory condition is a subgroup of patients with a significantly worse response to ICI. Through a case study, they identify neutrophils as an important mediator for immune suppression.

Will the work be of significance to the field and related fields? How does it compare to the established literature? If the work is not original, please provide relevant references. The work is in line with the established literature and builds on already published results concerning NLR and the combinatorial benefit of CTLA-4 and PD-L1 inhibition for PD-L1 therapy resistant patients.

The claim that inflammatory conditions are the cause of resistance to MSI-H CRC ICI resistance is new and of interest to the field.

Does the work support the conclusions and claims, or is additional evidence needed?

For their primary claim the authors use a cohort of 73 patients treated with ICI. The patients are primarily MSI, but approx. 30 % are either MSS or undetermined dMMR status, which may influence the data. In addition, 50 % received ICI monotherapy and 50 % did not. The authors should state more clearly which treatment the patients received instead (table 1). It would be of great benefit to the manuscript if the authors could validate their findings in another ICI treated cohort.

Response: Thank you for your advice. We have removed the MSI-L/MSS and undetermined dMMR cases, and we have also included new MSI-H cases. The latest cohort includes 62 MSI-H CRC patients (Line 87-89, Table 1). Those are all the cases who meet the inclusion criteria in our center.

In figure 1D-E, the authors use organoids from three patients with different response to treatment eg. PD, SD and CR. To fully support their claim about local immune suppression within tumors from patients with inflammatory conditions and differential ability to activate T-cells mimicking treatment response, the authors need to include more patient-organoid and immune cells for each condition.

Response: Thank you for your constructive advice. The latest version of the study includes organoids from 4 patients (two of them had PD with inflammatory conditions, one had TRG 3 and one had TRG 2 without inflammatory conditions, Figure 1D-F, Supplementary Figure 2).

The findings from scRNA are based on one patient, and there is no data from patients without an inflammatory condition to compare with. To fully support their claim, the authors should include more patients with inflammatory conditions as well as patients without inflammatory conditions in the scRNA analysis.

Response: Thank you for your constructive advice. The latest version of the study includes scRNA sequencing from 4 patient (Figure 2-3, Supplementary Figure 3-7), among whom 2 had inflammatory conditions and the rest had no inflammation.

Are there any flaws in the data analysis, interpretation and conclusions? - Do these prohibit publication or require revision?

In figure 2F the authors investigate the effect of a-CTLA4 and neutrophils on T-cell mediated cancer cell killing. They observe an increased apoptosis rate of tumor cells upon addition of CTLA4.

Preincubation of T-cells with neutrophils gives a slight decrease in tumor-killing, which is abrogated by addition of CTLA4. The authors claim that this supports the hypothesis of neutrophils inhibiting the T-cells through CD86-CTLA4 interaction. The change in apoptosis is, however, also present without neutrophils, indicating that it is a matter of an a-CTLA4 effect on T-cells independent of neutrophils.

Since the hypothesis of neutrophils inducing immune suppression through CD86-CTLA4 interaction is based on scRNA analysis from only one tumor and the link between neutrophil-CD86 and CTLA-4 induced inhibition is not clearly made, the conclusion needs more data to be substantiated.

Response: Thank you for your suggestion. We have re-done the experiment by neutralizing the CD80 and C86 in neutrophil instead of CTLA-4 in T cells (Figure 3E). Besides, we have change the murine tumor cells to be MC38, and T cells as well as neutrophils were derived from C57BL/6 mice.

In the conclusion (L235) the authors suggest adding a-CTLA4 for a better ICI sensitivity in MSI-H CRC patients with inflammatory conditions, but since approx. 50 % of the 73-patient cohort did not receive monotherapy and may hence have already received combination therapy together with the unknown treatment regime for the TCGA cohort, the authors should clarify this point and consider revising the conclusion if not in line.

Response: Thank you for your constructive advice. Among the latest inclusion of 62 patient, none of them received a-CTLA4 therapy. Also, we revised the baseline characteristic to be “PD-1 blockade combined with non-ICI treatment” (Table 1). We apologize for our poor description.

Is the methodology sound? Does the work meet the expected standards in your field?

The methods used in the study are sound. The authors include both clinical as well as translational data to support their study.

The in vitro data using organoids is, however, based on few/single samples which makes the conclusions too vulnerable, which also goes for the scRNA analysis. To meet the standards of the field more samples will need to be included.

Response: Thank you for your constructive advice. We have included organoids from 4 patients and scRNA seq data of 4 patients (Figure 1 D-F, Figure 2-3, and Supplementary Figure 3-7).

For the organoid cultures, the authors should include how the organoids were validated eg. did they do SNP analysis or similar to check the match with the donor patient. Were the organoids cryopreserved and how?

Response: Thank you for your constructive advice. We did not conduct genetic validation of the organoids previously. After receiving your comment, we tried acquiring tumor specimen of those 4 patients from repository. However, specimens from Patient 2 and 3 are unavailable. Therefore, we only conducted pathological determination (Figure 1D). For cryopreservation, Organoid were digested into single cells using TrypLE Express (Gibco) and cryopreserved at -80°C in FBS containing 10% dimethyl sulfoxide (DMSO) (Line 312-313).

Figure 1E: the authors should include controls of organoids alone to see the background apoptotic signal for each organoid line.

Response: Thank you for your suggestion. The experiments have been conducted (Supplementary Figure 2)

Is there enough detail provided in the methods for the work to be reproduced?

The method section is clearly written. For the work to be reproduced, it would be beneficial if the number of cells used for in vitro experiments was stated. Also, the concentration of some reagents fx a-CD3 and a-CD28 (L294) are missing and some manufacturers fx Annexin V Apoptosis Detection Kit (L302). For the flow cytometry analysis, the authors should include their gating strategy, preferably as a supplementary figure. The statistics section is comprehensive, but it would be beneficial if the statistical methods used was stated in the figure legends.

Response: Thank you for your constructive advice. We have provided the concentration of the antibodies and the information of apoptosis assay (Line 325, 331-333, 341-342, 348-350). The gating strategies have been displayed (Figure 1E, 3E, and Supplementary Figure 2). Also, the statistical methods used have been stated in the figure legends.

Further comments:

Abstract: the abstract would benefit from a clearer statement of the specific research question for this paper.

Response: Thank you for your constructive advice. We have revised the abstract.

L59: In the introduction the authors state that inflammatory conditions are common complications in CRCs. It would be nice if the authors would support this with a reference.

Response: We have added the reference (Decker, K. M. et al. 2020. Line 51-52).

L111: the authors mention a supplementary table. I have not been able to find this among the documents.

Response: We apologize for our carelessness. The table has been attached.

L87: the MSI-H cohort. The authors describe the cohort as being an MSI-H cohort albeit 30 % of the patients are not MSI-H. This may be a bit confusing. The authors should

consider rephrasing.

Response: We have excluded the MSI-L/MSS and undetermined dMMR cases, and we have also included new MSI-H cases. The latest cohort includes 62 MSI-H CRC patients (Table 1).

L140: the TCGA cohort. If possible, the authors should include information about the ICI treatments used for this cohort.

Response: We apologize for the poor description. What we displayed is the predicted tumor response to ICIs based on analysis of RNA seq data. We have revised the description (Line 136-139).

Figure 1A: please use arrows to highlight the tumor/metastases

Response: We have added the arrows (Figure 1A).

Overall assessment:

The research question is exciting and interesting and the methodology in the manuscript is just and embarks on highly relevant techniques. I do however believe that more data is needed to draw the conclusions of the manuscript and support the claims before publication. I would encourage the authors too include some or all of the suggestions as stated about in order to do so.

Reviewer #3 (Remarks to the Author):

The authors explore the relation between inflammatory conditions in MSI-H CRCs and resistance to ICIs through neutrophil-associated immunosuppression. I will focus on some aspects of the paper.

On Line 88: " We found that inflammatory conditions during ICI treatment were correlated with a higher ratio of stable disease (SD) and PD (87.5% vs. 33.3%, $P < 0.001$) and worse progression-free survival (PFS) ($P = 0.001$) (Figure 1B and C)."

It is unclear the test performed and how. Please explain.

Response: Thank you. The Wald chi-square test (Figure 1B) and Kaplan-Meier method with Log-rank test (Figure 1C) were used. We have added the information in the figure legend. We apologize for the lack of description.

On Line 385: "Student's t test was used for the comparison of two sets of quantitative data that deviated from the Gaussian distribution. The Mann-Whitney U rank-sum test was used for the comparison of two sets of quantitative data that did not deviate from the Gaussian distribution. " This statement seems incorrect, I guess you switched the non deviated with deviated. How did you measure the deviation? Please explain.

Response: Thank you. We have revised the writing to be "Student's t test was used for the comparison of two sets of quantitative data that subject to the Gaussian distribution. The Mann-Whitney U rank-sum test was used for the comparison of

two sets of quantitative data that deviate from the Gaussian distribution.” The deviation was measured using Shapiro-Wilk test (Line 431-436).

On line 327: "cells with fewer than 500 or more 328 than 10,000 genes were excluded". Can you explain these upper and lower bound? Particularly relevant is why you excluded the cells with more than 10.000 expressed genes.

Response: In single-cell RNA sequencing, cells with too low or too high genes may represent low quality cells or contaminated cells. Cells with more than 10,000 expressed genes were removed from the dataset as they likely represent the contamination during sample processing. In order to improve the data quality, many articles adopt similar strategies to filter out potential pollution and low-quality cells (PMID: 32286271, 32302573, 32783921, 31892341).

On line 396: "The Pearson or Spearman rank correlation test was used to measure the relationship between two variables" please specify which did you use and when.

Response: Thank you. We have added the information in the figure legend (Figure 2E-F).

On line 397: " All P values were two-sided" This sentence is incorrect. Hypothesis tests can be two sided, not the p-values.

Response: Thank you. We have revised the writing (Line 442-443).

I would like to have more details on the data pre-processing and on the clustering approaches. A comparison of at least two approaches would be nice.

Response: The Patient 1 had three sequencing libraries. First, SeuratObjects were constructed based on the expression matrix of each library and genes expressed in less than 3 cells were removed. Then, the “PercentageFeatureSet” function was used to count the percentage of mitochondrial genes in each library. The parameters "nFeature_RNA > 500 & nFeature_RNA < 10000 & percent.Mt < 20" were used to filter low-quality cells or potential contamination (PMID: 32286271, 32302573, 32783921, 31892341). Next, the three libraries were merged and batch effects were examined using the “Dimplot” visualization method. The merged data were normalized and scaled using “SCTransform” function with the parameter “vars.to.regress = "percent.mt"”. The first 30 PCs were used to construct the SNN network, and the graph-based clustering method Louvain algorithm was used to identify the cell clusters with a resolution of 0.6. Finally, UMAP is used to visualize the clustering results in two-dimensional space. We use Seurat for the processing of single cell RNA-seq data because it is currently the most mainstream software package for single cell RNA-seq analysis, with over 14,700 citations (<https://www.scrna-tools.org/table>), and its accuracy has been widely confirmed by various studies.

REVIEWERS' COMMENTS

Reviewer #1 (Remarks to the Author):

The revised version of the manuscript is improved by incorporating new data (especially additional patient scRNA-seq data) and further analysis. I still have concerns about the following points.

1. The authors suggested that local inflammation or an elevated NLR ratio is associated with a poor ICI response in MSI-H CRC patients. The definition of "local inflammation" is not yet clear. In patients with tumor obstruction, perforation, or peritonitis, almost all will have significant systemic inflammation. If they suffered from systemic inflammation, why is only the local tumor infiltrated with neutrophils, but not in metastatic tumors? Whether antibiotic treatment resolves inflammation? Furthermore, the sampling time to evaluate systemic inflammation should be consistent for all cases, since there are too many factors that can influence the results of the blood test.

2. The study used scRNA data to indicate infiltrated immunosuppressive neutrophils in tumor samples. However, the number of cases is limited. I suggest performing IHC for neutrophil staining to directly visualize neutrophils in CRC samples in a larger cohort. The data will be important to confirm the influence of immunosuppressive neutrophils in CRCs receiving ICB treatment.

Reviewer #2 (Remarks to the Author):

The authors have to a great extent included the suggestions. The additional experiments have supported the overall relevance of the Treg:Myeloid CD86:CTLA4 axis and specifically the benefit of inhibiting this pathway in a setting of neutrophils and T-cells, which was supported by in vitro experiments in a murine system.

The additional in vitro experiments have, however, not supported the claim that this is of specific importance during inflammatory conditions and this claim still relies on the initial data of patient 1. Although a second patient with inflammation was included (patient 2) the data from this patient does not support the claim – possibly because the patient received timely anti-infectious treatment

RESPONSE TO REFEREES

Reviewer #1 (Remarks to the Author):

The revised version of the manuscript is improved by incorporating new data (especially additional patient scRNA-seq data) and further analysis. I still have concerns about the following points.

1. The authors suggested that local inflammation or an elevated NLR ratio is associated with a poor ICI response in MSI-H CRC patients. The definition of "local inflammation" is not yet clear. In patients with tumor obstruction, perforation, or peritonitis, almost all will have significant systemic inflammation. If they suffered from systemic inflammation, why is only the local tumor infiltrated with neutrophils, but not in metastatic tumors? Whether antibiotic treatment resolves inflammation? Furthermore, the sampling time to evaluate systemic inflammation should be consistent for all cases, since there are too many factors that can influence the results of the blood test.
2. The study used scRNA data to indicate infiltrated immunosuppressive neutrophils in tumor samples. However, the number of cases is limited. I suggest performing IHC for neutrophil staining to directly visualize neutrophils in CRC samples in a larger cohort. The data will be important to confirm the influence of immunosuppressive neutrophils in CRCs receiving ICB treatment.

Response: 1. Thank you. In our point, the local inflammation is caused mostly by infectious disease in colorectal cancer. Although gut microbiota influences both local and systemic inflammation chronically, it could still cause acute inflammatory reaction when exposed directly to immune system due to obstruction or perforation. We consider that the acute inflammatory reaction could stimulate the cytokine production and recruit the inflammatory cells systemically. Meanwhile, despite that the immune status of metastatic site could be influenced by circulating cytokine, it is still less likely to participate in the recruitment of inflammatory cells. According to the study of Wang, et al, tumor-infiltrating neutrophils exhibited an activated phenotype compared with normal activated peripheral neutrophils. Therefore, although suffered from systemic inflammation, the metastatic site could have less neutrophil infiltration and better immune status than primary tumor with local inflammation.

As for the sampling, our point is that NLR itself, instead of systemic inflammation, could be a predictor to poor response to PD-1 blockade. Existing studies demonstrated a positive correlation between NLR and tumor neutrophil infiltration, while an elevated NLR could attribute to both inflammation and non-inflammatory situations. Thank you for your constructive comment, we have revised the grouping and

re-analyzed the data (Line202-207, 301-302 Figure 4B to D).

2. Thank you, we have performed the IHC staining for neutrophils and analyzed the result (Line135-138, Supplementary Figure 7).

Reviewer #2 (Remarks to the Author):

The authors have to a great extent included the suggestions. The additional experiments have supported the overall relevance of the Treg:Myeloid CD86:CTLA4 axis and specifically the benefit of inhibiting this pathway in a setting of neutrophils and T-cells, which was supported by in vitro experiments in a murine system.

The additional in vitro experiments have, however, not supported the claim that this is of specific importance during inflammatory conditions and this claim still relies on the initial data of patient 1. Although a second patient with inflammation was included (patient 2) the data from this patient does not support the claim – possibly because the patient received timely anti-infectious treatment.

Response: Thank you for your constructive comment. To further demonstrate the association between the neutrophil infiltration and the L-R pair interaction, we investigate the correlation between *ITGAM*, the marker of activated neutrophil, and markers with significant interactions in the current study in TCGA. The analysis indicate that *ITGAM* expression was positively correlated to the expression of *CD80*, *CD86*, *CTLA4*, *CSF1*, *CSF1R*, *CCL5*, *CCR1*, *CCR5*, and *CXCL9* (Line174-180, Supplementary Figure10). This phenomenon further supports that elevated neutrophil infiltration correlates to activated CD80/CD86-CTLA4 axis.